# Establishment and Application of a Patient-Ventilator Asynchrony Remote Network Platform for ICU Mechanical Ventilation: A Retrospective Study

**DOI:** 10.3390/jcm12041570

**Published:** 2023-02-16

**Authors:** Longxiang Su, Yunping Lan, Yi Chi, Fuhong Cai, Zhenfeng Bai, Xianlong Liu, Xiaobo Huang, Song Zhang, Yun Long

**Affiliations:** 1Department of Critical Care Medicine, State Key Laboratory of Complex Severe and Rare Diseases, Peking Union Medical College Hospital, Chinese Academy of Medical Science and Peking Union Medical College, Beijing 100730, China; 2Intensive Care Unit, Sichuan Academy of Medical Sciences & Sichuan Provincial People’s Hospital, School of Medicine, University of Electronic Science and Technology, Chengdu 610000, China; 3Shanghai Shumu Medical Technology Co., Ltd., Shanghai 201103, China; 4Center for Medical Device Evaluation, National Medical Products Administration, Beijing 100081, China

**Keywords:** patient-ventilator asynchrony, ICU mechanical ventilation, remote network platform, double triggering, ineffective triggering

## Abstract

Background: In the process of mechanical ventilation, the problem of patient-ventilator asynchrony (PVA) is faced. This study proposes a self-developed remote mechanical ventilation visualization network system to solve the PVA problem. Method: The algorithm model proposed in this study builds a remote network platform and achieves good results in the identification of ineffective triggering and double triggering abnormalities in mechanical ventilation. Result: The algorithm has a sensitivity recognition rate of 79.89% and a specificity of 94.37%. The sensitivity recognition rate of the trigger anomaly algorithm was as high as 67.17%, and the specificity was 99.92%. Conclusions: The asynchrony index was defined to monitor the patient’s PVA. The system analyzes real-time transmission of respiratory data, identifies double triggering, ineffective triggering, and other anomalies through the constructed algorithm model, and outputs abnormal alarms, data analysis reports, and data visualizations to assist or guide physicians in handling abnormalities, which is expected to improve patients’ breathing conditions and prognosis.

## 1. Introduction

Mechanical ventilation (MV), which more than one-third of patients in the intensive care unit (ICU) receive, is the most important form of respiratory support for critically ill patients [1]. Patient-ventilator asynchrony (PVA), defined as a mismatch between demands of the patient respiratory system and the ventilator, is very common during mechanical ventilation [2,3]. PVA often brings a series of adverse effects, including discomfort, air hunger, increased respiratory effort, muscle injury, decreased sleep quality, and increased sedation or muscle relaxant demands, which can lead to aggravation of ventilator-related lung injury, prolonged mechanical ventilation, difficulty in weaning, and even increased mortality [4,5,6,7]. Timely recognition and treatment of PVA will improve patient comfort and, potentially, outcomes. The incidence of PVA may be largely underestimated due to inadequate monitoring or inexperience of physicians [8]. To improve the recognition of PVA, we developed a remote mechanical ventilation visualization network system and, simultaneously, a related automatic recognition algorithm for PVA. The system can accurately and quickly identify several types of PVA in real time, provide real-time feedback through the cloud platform, and, finally, form a report of mechanical ventilation for clinical reference.

## 2. Materials and Methods

### 2.1. Design Framework

Aiming at the above problems, this study proposes a remote monitoring (Remote-VentilateView) platform for ICU mechanical ventilation. The Remote-VentilateView platform architecture is mainly divided into three layers: the data source layer, the data processing layer, and the data application layer, as shown in Figure 1. The data source layer, located at the base of the architecture, is the data source of the overall architecture and is responsible for importing, desensitizing, encrypting, compressing, and forwarding the original data of the ventilator. The data processing layer is the core layer of the overall architecture and the key to embodying data intelligence. This layer mainly uses cloud computing for big data processing of massive ventilator data. The data on the ventilator are summarized, sorted, stored, parsed, and restored on the cloud platform. After that, data will be cleaned, segmented, and aggregated. The data application layer is the link that directly reflects the value and application side of the data. It can aggregate a variety of ventilator data for centralized visual display. It is a comprehensive evaluation of long- and short-term data to generate a dynamic detection report.

The specific process is that the real-time data generated by the patient are transmitted to the Remote-VentilateView platform server through the ventilator, Data Transfer Unit (DTU, Jinan Usr IOT Technology Limited), switch, desensitization encryption server, and Virtual Private Network (VPN). To improve the security of data transmission, ventilator data are transmitted through the VPN and provide security protection capabilities through security gateways, firewalls, intrusion detection and prevention systems (IDPS), and other equipment. The Remote-VentilateView platform can store data and perform data classification, data analysis, and other processing, providing data support for event alarms, data analysis reports, data visualization and other data applications.

This study takes severe mechanical ventilation as a practical application scenario and uses the Remote-VentilateView platform to practically solve the most common patient-ventilator asynchrony problem in mechanical ventilation—the identification and alarm of double triggering and ineffective triggering.

### 2.2. Software and Hardware Equipment

The ventilator models involved in this study include Mindray-SV300 (Shenzhen Mindray Bio-Medical Electronics Co., Ltd.), Mindray-SV800 (Shenzhen Mindray Bio-Medical Electronics Co., Ltd.), Vyaire-TBird (Vyaire Medical Products (Shanghai) Co., Ltd.), Covidien Puritan Benett 840 (Covidien (China) Medical Devices Technology Co., Ltd.), Maquet Servo-s (Maquet (Shanghai) Medical Equipment Co., Ltd.), and other common ventilators on the market. The pressure (Paw)/time waveform, flow rate (flow)/time waveform, volume (volume)/time waveform, etc., displayed by the ventilator are stored in the database as raw data. The sampling frequency of data is 50 Hz, the unit of Paw is cmH_2_O, the unit of flow is L/min, and the unit of volume is mL. The DTU involved in this study is the USR IOT-G771/G781, and the data transmission method adopts 4G wireless transmission communication and RS232 communication. The front-end PC involved in this research is configured as CPU: 4 cores and 8 threads, main frequency 2.8 GHz; memory: 16 G; hard disk: 256 G. The operating systems involved in this study are CentOs-7.5 and Ubuntu-18.04.

Alibaba/China Mobile Cloud is the cloud computing service provider involved in this study. Other cloud computing services include MySQL, TableStore/Hbase, Nginx, Redis, Object Storage Service, Kafka, MQTT, firewall, load balancing, etc. The development languages involved in this research are Java-1.8, Python3.8, and TypeScript-3.9, and the components include OpenVPN-2.5, OpenSSL-3.0, Spring Cloud-1.0, Spring Boot-2.2, Vue-2.0, Numpy-1.23, Pandas-1.5, etc.

### 2.3. Data Source Processing and Data Labeling

We extracted historical data or real-time data, and performed data preprocessing, including, but not limited to, data normalization (Z-score normalization, Min-max normalization, normalization, etc.), data cleaning (data smoothing, denoising, etc.), and data labeling. The purpose of data normalization is to eliminate the differences between features to facilitate data analysis and comparison and balance the weights learned by algorithms. Data cleaning filters the noise of original data to obtain higher-quality data. Data noise affects the accuracy of the algorithm recognition results.

The purpose of data annotation is to help machine learning algorithms to learn data features. At present, most of the data are raw data without labels; to obtain labeled data, some data need to be labeled. The data and the corresponding labels constitute a labeled dataset, which can be used for model training and model evaluation of supervised learning algorithms.

Users can obtain historical data from the annotation service of the Remote-VentilateView platform every day and visualize it with optional lengths such as 1 min and 3 min. Respiratory specialists manually annotate the data, and annotations are divided into point annotation, event annotation, and whole segment annotation, which correspond to the patient-ventilator asynchrony annotation corresponding to the single time point, the start and end of the event, and the whole annotation of a segment. The annotation type supports ineffective triggering, double triggering, and other types of patient-ventilator asynchrony, such as delayed switching and flow insufficient.

### 2.4. Construction of the Remote Network

Each piece of hospital equipment is connected to the Remote-VentilateView platform through a data communication link, which consists of three parts: business cluster, communication cluster, and data support cluster. Data desensitization refers to the transformation of some sensitive information through desensitization rules to achieve reliable protection of sensitive private data. Data encryption refers to the transformation of plaintext into ciphertext through encryption algorithms and encryption keys. It is a reliable method for computer systems to protect information. In view of the protection of data privacy, the platform will desensitize and encrypt sensitive patient data to maximize the protection of patient privacy. The Remote-VentilateView platform is connected to massive data devices or databases and can perform analysis and processing in real time, discover high-risk data, issue event alarms, remind medical staff to intervene quickly, and automatically generate electronic data analysis reports and visualize real-time respiratory waveforms. The Remote-VentilateView platform reminds clinicians in real time through the alarm function, strives for early identification of abnormal patient-ventilator asynchrony events, and optimizes mechanical ventilation treatment.

### 2.5. Statistical Methods

Sensitivity, also known as the true positive rate, is the ability of the model algorithm to identify abnormal data. The higher the sensitivity, the lower the probability of missed diagnosis. Specificity, also known as the true negative rate, is the probability that the model algorithm recognizes “normal” data in normal data. The higher the specificity, the lower the misdiagnosis rate. Positive predictive value is the probability that the model is correct in predicting all outcomes for which the predicted outcome is positive. Negative predictive value is the probability that the model predicts incorrectly for all outcomes for which the predicted outcome is negative. This study mainly focused on ineffective triggering and double triggering. In this study, manual annotation is used as the gold standard for model training and model validation, and the results of the recognition algorithm are compared with manual annotation.

## 3. Result

### 3.1. Data Source Layer Construction

The equipment mainly involved in this study includes ventilators, routers, DTUs, network cables, switches, gateways, servers, etc. The main communication methods used by the ventilator of this platform are serial and network communication.

As shown in Figure 2, the ventilator is connected to the switch or router through wired transmission, WIFI, mobile network, and other wireless transmissions through DTU, network cable direct connection, serial port to network cable, etc. At present, most ventilators have a real-time transmission function. In this case, the ventilator can use the wired network in the hospital to transmit data to the Remote-VentilateView platform in real time. Traditional ventilators do not support real-time transmission. In this case, you need to connect the ventilator through serial communication by DTU and use DTU’s wired and wireless connection to the superior switch and router to realize the Remote-VentilateView platform connection.

The ventilator data transmitted by DTU can be sent to the Remote-VentilateView platform by the front-end machine or the server in the hospital via the network. As a network conversion device, the front-end computer is managed by the hospital and can be used to deploy visualization monitoring, event alarms, dynamic reports, and other intranet application services separately in the hospital network. Data desensitization services shall be deployed on front-end computers and hospital servers to avoid exposing patient privacy, and then security components, such as VPN and firewall, to prevent potential security risks such as data leakage and data tampering will be deployed.

### 3.2. Data Processing Layer

The data processing layer relies on the distributed CAP (Consistency, Availability, Partition Tolerance) theory and uses stream processing, Subscribers/Publishers (Sub/Pub) services, MapReduce, consistency/locality-sensitive hash, and other technologies to analyze massive ventilator waveforms. Data analysis, data standardization, data cleaning, data labeling, event alarms, data analysis reports, and data visualization are output through the algorithm model, as shown in Figure 3. The process of data standardization is very important because different brands of ventilators often have different data formats. After parsing and decompressing the datagram transmitted over the network, the ventilator data waveform is reconstructed. Data standardization is used to unify the features to better train a machine learning algorithm. Data cleaning is performed using machine learning methods such as range constraints, mean/median filling, and anomaly detection. Machine learning algorithms have better learning effects on labeled data, but the original data of ventilators are unlabeled, so this study professionally labeled certain data. The processed data are distributed and converted according to actual business needs, provided to doctors, nurses, and analysts, and displayed in the form of event alarms, data analysis reports, visualization, etc. A feedback loop is provided at the same time, allowing us to save, modify, and mark the history of event alarm and ventilator waveform/numerical parameters. Data export, scientific research software import, etc., are provided for scientific research and the experimental needs of mechanical ventilation.

The patient-ventilator asynchrony anomalies we focus on are double triggering and ineffective triggering. The data in the database include created time, Paw, flow, volume, ventilation mode, PEEP configuration, etc. Among them, Paw, flow, and volume are waveform data related to time. Some parts of data fed into the algorithm model are the main dependent data for judging the event alarm.

The discriminant rule for double triggering is that the volume–time curve falls vertically to the baseline; there is no expiratory waveform between two ventilations, or there is an expiratory phase, and the expiratory time is less than half of the average inspiratory time.

The judgment rule for ineffective triggering is to identify the “flow rate deflection” characteristic of the expiratory flow velocity curve that first rises and then falls by calculating the first derivative function curve of the expiratory flow–time curve and then determining the local maximum and minimum values. The time position of the value corresponds to the flow rate value. When the amplitude (maximum value–minimum value) of “flow velocity deflection” is greater than the set threshold (5 L/min) and the time of “flow velocity deflection” (the time interval between the maximum value and the minimum value) is greater than the preset value threshold (0.12 s), the algorithm recognizes this “flow rate deflection” as an ineffective triggering event.

### 3.3. Patient Identification

As shown in Figure 4, (a) is a typical waveform diagram of double triggering, and (b) is a typical waveform diagram of ineffective triggering. Double triggering refers to two consecutive ventilator cycles with an expiratory time less than half of the average inspiratory time [2]. Double triggering is characterized by incomplete exhalation between breaths, followed by a rapid increase in the second flow rate after the initial trigger, and the expiratory time is less than half of the average inspiratory time [2]. Ineffective triggering refers to the patient’s inhalation. Inspiratory effort fails to trigger ventilator delivery [9]. Ineffective triggering is manifested as ineffective inspiratory effort in the expiratory phase, positive changes in flow, and negative changes in pressure [9].

This study involved four patients’ ventilator waveforms, which contained a total monitoring time as high as 1284.39 h (Table 1). The two anomalies involved in this study are double triggering and ineffective triggering. The accuracy results are shown in Table 2, with a total of 4496 breaths, of which 716 were double triggering events. The experimental results of the ineffective triggering algorithm are shown in Table 3, with a total of 4496 breaths, of which 910 were ineffective triggering events.

Among the 4496 breaths, there were 716 actual positive events and 3780 actual negative events of double triggering, and 484 predicted positive events and 4012 predicted negative events for the double triggering algorithm. According to the experimental results, there were 481 true positives, 3 false positives, 235 false negatives, and 3777 true negatives. From this calculation, sensitivity = 481/716 = 67.18%, specificity Sp = 3777/3780 = 99.92%, positive predictive value PPV = 481/484 = 99.38%, and negative predictive value NPV = 3777/4012 = 94.14%.

Among the 4496 breaths, there were 910 actual positive events and 3586 actual negative events of ineffective triggering, and 929 predicted positive events and 3567 predicted negative events for the ineffective triggering algorithm. According to the experimental results, there were 727 true positives, 202 false positives, 183 false negatives, and 3384 true negatives. From this calculation, sensitivity Sv = 727/910 = 79.89%, specificity Sp = 3384/3586 = 94.37%, positive predictive value PPV = 727/929 = 78.26%, and negative predictive value NPV = 3384/3567 = 94.87%.

At present, the sensitivity recognition rate of the ineffective triggering algorithm is 79.89%, and the sensitivity recognition rate probability of the double triggering algorithm is 67.17%. Among the two types of anomaly detection, the algorithm for ineffective triggering has a higher recognition sensitivity and recognition rate, which is 12.72% higher than the recognition sensitivity rate of the double triggering algorithm.

### 3.4. Data Analysis Report

In the data application layer, the data analysis report is one of the important applications. The data analysis report shown in Figure 5 includes basic information, diagnostic reference, prompts and findings, parameters of the mechanical ventilation ventilator, and the respiratory waveform. The basic information is for doctors to identify the patient’s identity and clarify the hardware parameters of the ventilator, including the basic information of the patient and the hospitalization information of the patient. The diagnostic reference gives the reason for the alarm, where the severity of patient-ventilator asynchrony is represented by the asynchrony index, which (expressed as a percentage) = number of asynchrony events/total respiratory rate (number of ventilator cycles + ineffective trigger times) × 100% [2,10,11]. The patient-ventilator asynchrony index is calculated on the patient’s ventilator data in real time. The sliding time window is 3–6 min, and the sliding step is 1 s. Severe patient-ventilator asynchrony is defined as an asynchrony index ≥ 10%. Within the sliding window range, when the patient-ventilator asynchrony index is greater than the threshold, an abnormal event warning will be performed, and the data analysis report will be generated and pushed to the clinic. As shown in the report in Figure 5, the asynchrony index reached 13% in 4 min. The real-time monitoring report will be generated immediately and pushed to the clinic. Prompts and findings are used to display the analysis, description, and conclusions of the mechanical ventilation treatment data and will give treatment suggestions to the alarm, which need to be signed and confirmed by the doctor. The parameters of mechanical ventilation ventilators include peak pressure, positive end-expiratory pressure, total respiratory rate, tidal volume, minute ventilation, oxygen concentration, etc. A respiratory waveform graph is a detailed analysis report showing one or more mechanical ventilation treatment data in the form of graphs, including a pressure–time graph, flow–time graph, volume–time graph, volume–pressure graph, and flow–volume graph, according to different anomalies showing different characteristics on the waveform, to judge whether these waveforms are normal or not.

## 4. Discussion

In this study, we established Remote-VentilateView, an intelligent, efficient, and visualized real-time alarm monitoring system for mechanical ventilation driven by big data technology and algorithms. The system provides monitoring of mechanical ventilation parameters and waveforms, PVA event alarming, and dynamic analysis reports to improve the safety of mechanical ventilation. Through internal validation, our system showed relatively high accuracy in identifying double triggering and ineffective triggering.

Mechanical ventilation is the most common respiratory support in the ICU [12]. Intensivists could choose fully ventilator-controlled modes, relieving the patient’s respiratory load or, partially assisted mode, allowing the existence of breathing effort, according to the severity and duration of lung injury [13]. Perfect patient-ventilator interaction should minimize excessive respiratory load and retain moderate spontaneous breathing effort to prevent atrophy of the diaphragm muscle. However, this balance is fragile, and it is difficult to avoid PVA during the process [14]. PVA refers to the phenomenon of incoordination between the patient and the ventilator caused by the mismatch between the patient’s neural inspiratory time and the ventilator’s inspiratory time or the mismatch between the ventilator support and demand during the entire respiratory cycle [15,16,17]. Data showed that the incidence of PVA varied between 10% and 85% [8]. The reported broad range of incidence rate was due to differences regarding the definition of PVA, physician experience, observation timing, etc. [11,18]. In clinical practice, PVA is usually identified based on a patient’s breathing state and the ventilator waveform. However, studies have shown that the accuracy of nonexpert intensivists in identifying PVA through the ventilator waveforms at the bedside was rather poor [3]. An international research study showed that participation in specific mechanical ventilation-related training and a training time of more than 100 h can improve the accuracy of physicians in identifying PVA [19]. Nonetheless, such training was not perfect and applicable everywhere. Esophageal pressure or electrical activity of the diaphragm monitoring does help in the recognition of PVA [3]. However, these are semi-invasive procedures that are quite complex to perform, requiring placement of a dedicated tip balloon catheter, which is mostly for research use [20]. Ramsay et al. found that, in 28 patients with obstructive and restrictive lung disease given noninvasive ventilation, the addition of parasternal electromyography significantly increased the recognition of PVA [21]. This study proposes to build a Remote-VentilateView platform to solve various PVA problems. This platform can partly solve the needs of doctors in critical care medicine for the use of mechanical ventilation and lays the foundation for large-scale clinical application in the later stage.

Various PVA algorithms select trending real-time waveform data and automatically combine the ventilation mode and other setting parameters to establish complete rule decision logic by deriving mathematical formulas, fitting curve characteristics, and cross-validation thresholds. We used mapping regulations of waveform performance and change with significant differences in different PVA events, and mining deeply regulations of waveform performance and change without significant difference in order to eliminate interference between events, therefore realizing long-term monitoring of respiratory waveforms and accurate recognizing of PVA events. Events are precisely identified. In this study, the algorithm, using expert experience combined with morphological observation, can achieve good results in the recognition of double triggering and ineffective triggering. By using the Remote-VentilateView platform and the intelligent algorithm to automatically label and accumulate data, a large amount of high-quality data can be established. In summary, there are few available products that have been released regarding machine learning algorithms for mechanical ventilation. Based on the foundation of our current algorithms, we can accumulate a large amount of labeled data and provide large samples for the research of machine learning algorithms. Our study integrates the experience of professionals in manual identification and develops relevant algorithms to automatically identify abnormal clinical situations including double triggering and ineffective triggering. At the same time, remote network and cloud platform technology are used to realize the tasks of continuous and dynamic real-time monitoring and alarming of these identification algorithms. It has the potential for large-scale clinical application.

In addition, using machine learning algorithms to identify and predict PVA is a feasible method to replace tedious manual labor and improve accuracy. By analyzing 4.26 million breaths in 62 mechanically ventilated patients at risk for or diagnosed with ARDS, Sottile et al. applied Python and SciPy scientific stack (an open-source programming and scientific analysis toolset that includes machine learning algorithms and a cross-sectional verification method) technology to develop algorithms to identify PVA [5]. Through this technology, a set of data is analyzed, and the characteristics of each breath are used to determine whether there are the following two types of PVA: double triggering and ineffective triggering. Then, we iteratively developed a machine-learning model to classify each breath as normal or asynchrony. The overall accuracy of the above two types of PVA identification was 91%, and the area under the ROC curve was 0.92. The accuracy and area under the ROC curve were 97% and 0.95, and 79% and 0.80 for the double triggering and ineffective triggering. In the verification experiment, the accuracy and ROC of double triggering and ineffective triggering have achieved the expected results. Although the accuracy of the machine learning algorithm in identifying PVA is relatively high from the data, it is still limited to specific types of PVA. For the types with less obvious waveform features, such as early or delayed cycling and delayed triggering, the accuracy needs to be further improved. The algorithms we developed for the two most common types of PVA have also shown high accuracy, reaching a level that can be used in clinical practice. However, its sensitivity still needs to be optimized.

The anomaly algorithm model provided in this study has the advantages of strong interpretability, adaptive feature matching, and fast algorithm identification. There are still the following shortcomings in this study: (1) the database used for training the model is currently small, and, if the algorithm recognition ability is enhanced, the database needs to be expanded; (2) more validation datasets are used to obtain better and more accurate thresholds for generalization ability; and (3) to dynamically adapt to the learning dataset, a design that dynamically adjusts the threshold according to the dataset can be added. At present, we are also continuously optimizing and adjusting the response recognition capabilities of related algorithms and platforms.

## 5. Conclusions

This study focuses on the problem of insufficient recognition of PVA in mechanical ventilation in critically ill patients. By building a remote network cloud platform for mechanical ventilation, the real-time transmission of the waveform data of the ventilator is realized and an artificial intelligence algorithm is constructed to automatically identify the types of double triggering and ineffective triggering. PVA is detected, and a real-time alarm is issued through the cloud platform to guide clinicians to identify and address PVA at an early stage to improve the prognosis of mechanically ventilated patients.

## Figures and Tables

**Figure 1 jcm-12-01570-f001:**
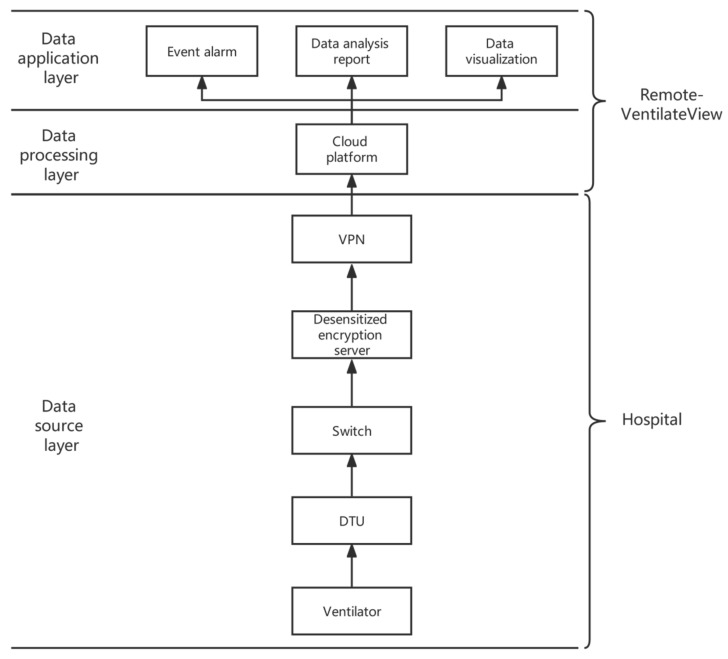
Remote-VentlateView platform architecture.

**Figure 2 jcm-12-01570-f002:**
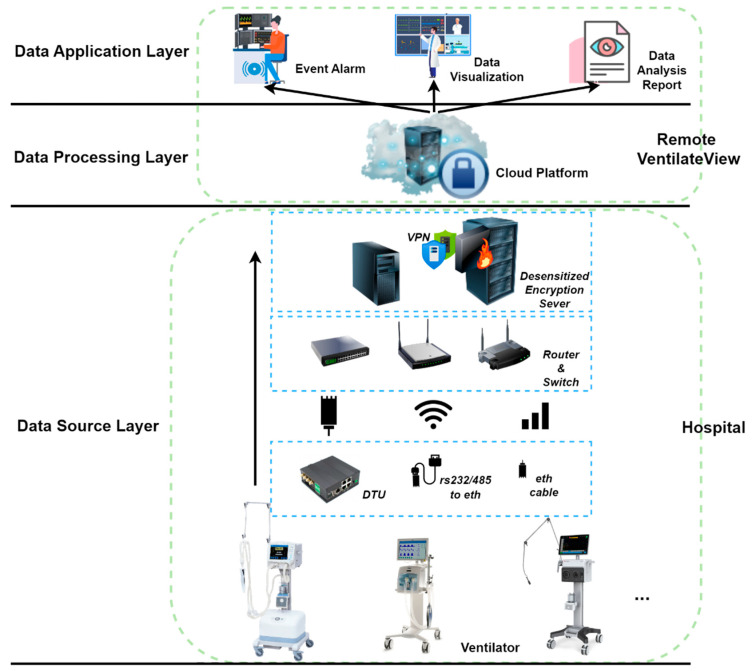
Remote-VentilateView network topology.

**Figure 3 jcm-12-01570-f003:**
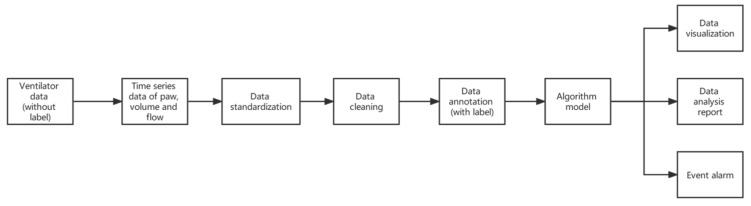
Flow chart of mechanical ventilation event alarm.

**Figure 4 jcm-12-01570-f004:**
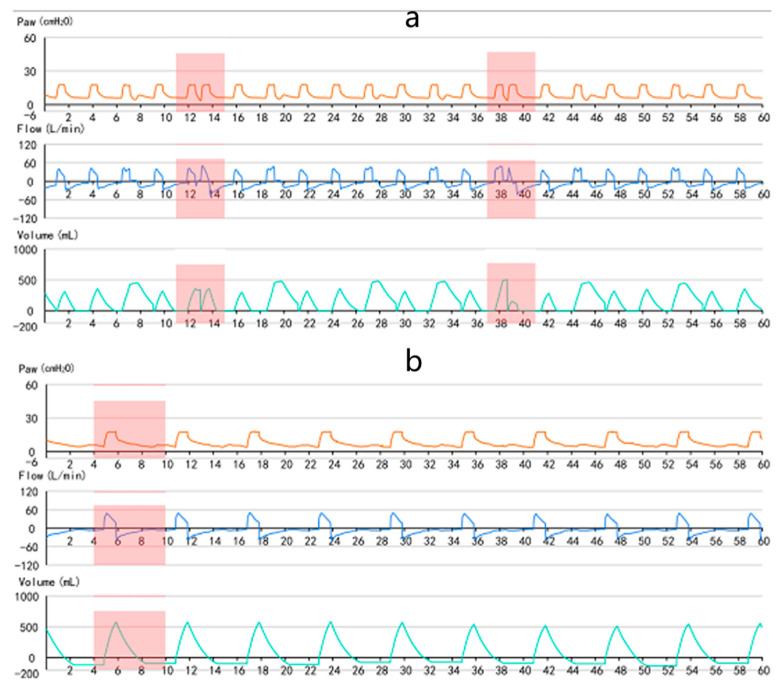
Data abnormal waveform. (**a**) Double triggering waveform. (**b**) Ineffective triggering waveform.

**Figure 5 jcm-12-01570-f005:**
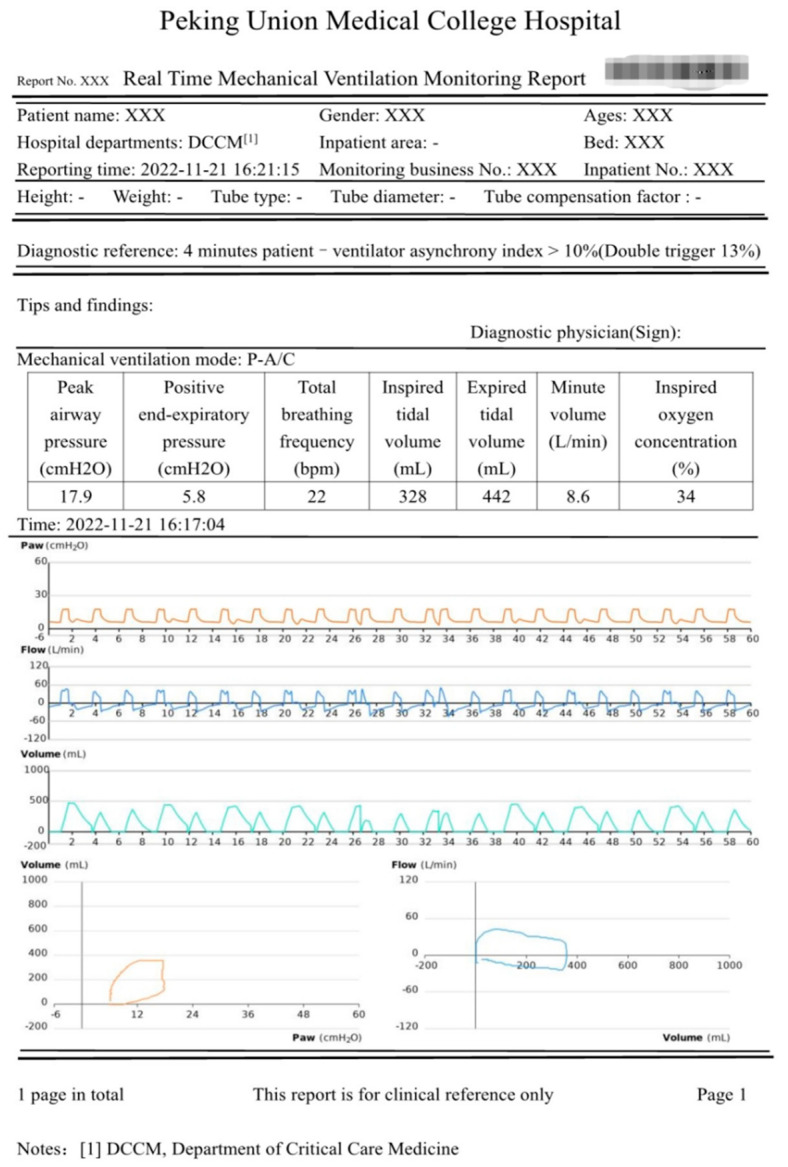
Double triggering data analysis report.

**Table 1 jcm-12-01570-t001:** Basic information and monitoring information of 4 patients.

Attribute	Value			
Gender	Male	Male	Male	Male
Age	84	62	70	63
Data duration (h)	880.12	115.97	66.6	221.7
Ventilation mode	CMV	CPAP/PSV	V-A/C	V-A/C
Tidal volume (mL)	440 ± 14.8	400 ± 14.8	420 ± 14.8	430 ± 14.8
Peak pressure (cmH_2_O)	-	19 ± 9.5	21 ± 9.5	40 ± 9.5
Mean airway pressure (cmH_2_O)	10 ± 2.2	8 ± 2.2	12 ± 2.2	14 ± 2.2
Fraction of inspiration O_2_ (%)	40 ± 26.0	40 ± 26.0	40 ± 26.0	100 ± 26.0
Positive end expiratory pressure (cmH_2_O)	-	5	5	5

**Table 2 jcm-12-01570-t002:** Experimental results of the double triggering algorithm.

	Gold Standard (Manual Marking Method)		
	Positive *n* = 716	Negative *n* = 3780	
positive *n* = 484	true positive A = 481	false positive B = 3	positive predictive value PPV = 99.38%
negative *n* = 4012	false negative C = 235	true negative D = 3777	negative predictive value NPV = 94.14%
	sensitivity Sv = 67.18%	specificity Sp = 99.92%	

**Table 3 jcm-12-01570-t003:** Experimental results of the ineffective trigger algorithm.

	Gold Standard (Manual Marking Method)		
	Positive *n* = 910	Negative *n* = 3586	
positive *n* = 929	true positive A = 727	false positive B = 202	positive predictive value PPV = 78.26%
negative *n* = 3567	false negative C = 183	true negative D = 3384	negative predictive value NPV = 94.87%
	sensitivity Sv = 79.89%	specificity Sp = 94.37%	

## Data Availability

The data that support the findings of this study are available from the corresponding author upon reasonable request.

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
