# Peer review of "Establishment and Application of a Patient-Ventilator Asynchrony Remote Network Platform for ICU Mechanical Ventilation: A Retrospective Study"

_jcm, 2023, doi:10.3390/jcm12041570_

Round 1

Reviewer 1 Report

This is an engaging manuscript focusing on the performance of an algorithm developed to detect ineffective triggering and double triggering.

The idea is of interest and relevant in the field. The paper is almost well written, some fine minor spell check can be improved.  

Data are clearly presented, moreover tables and figures are informative.   However some minor comments must be addressed in order to elicit publication of the manuscript.  

I suggest the authors to better specify the type of study they have conducted, maybe also in the title. They have derived results from 4496 breaths. Is this number enough wide to assume the obtained results are generalizable?

The setting of patients included and the number of them must be specified. A summary table on the characteristics of included patients would be appreciated, particularly the initial diagnosis of included patients is lacking.

Author Response

This is an engaging manuscript focusing on the performance of an algorithm developed to detect ineffective triggering and double triggering.

The idea is of interest and relevant in the field. The paper is almost well written, some fine minor spell check can be improved.  

Data are clearly presented, moreover tables and figures are informative.   However some minor comments must be addressed in order to elicit publication of the manuscript.  

Response:

Thank you very much for your general comments.

I suggest the authors to better specify the type of study they have conducted, maybe also in the title.

Response:

Thank you very much for your suggestion.

The study type of this article is retrospective study. We have added these information in the title.

They have derived results from 4496 breaths. Is this number enough wide to assume the obtained results are generalizable?

Response:

Thank you very much for your comments.

This article focuses on developing a platform, building a model, and then conducting small-scale verification. Although the number of clinical patients included is not large, the total time is long enough. Therefore, it is sufficient from the data used for model validation. Of course, in the follow-up research, we planned to continue to study the relationship between the safety and effectiveness of the single-center clinical verification system and the multi-center automatic human-machine asynchronous pattern recognition and patient prognosis, so as to lay a more solid foundation for clinical promotion.

The setting of patients included and the number of them must be specified. A summary table on the characteristics of included patients would be appreciated, particularly the initial diagnosis of included patients is lacking.

Response:

Thank you very much for your comments.

The model validation data in this paper involves 4 patients, and the total monitoring time is as high as 1284.39 hours. The specific information is shown in the table below.

Table 1 Basic information and monitoring information of 4 patients

Gender

Male

Male

Male

Male

Age

84

62

70

63

Data duration(h)

880.12

115.97

66.6

221.7

Data start and end

2021/11/30 18:38 -2021/12/27 6:00

2022/07/06 18:39 -2022/07/09 10:00

2022/07/20 16:24 -2022/07/22 10:00

2022/03/14 09:08 -2022/03/18 12:00

Ventilation mode

CMV

CPAP/PSV

V-A/C

V-A/C

Tidal volume(mL)

440±14.8

400±14.8

420±14.8

430±14.8

Peak pressure (cmH2O)

-

19±9.5

21±9.5

40±9.5

Mean airway pressure (cmH2O)

10±2.2

8±2.2

12±2.2

14±2.2

Fraction of inspiration O2(%)

40±26.0

40±26.0

40±26.0

100±26.0

Positive end expiratory pressure (cmH2O)

-

5

5

5

Reviewer 2 Report

Thank you for the opportunity to review this article.

This article promotes a network system with an automatic recognition algorithm that improve the accurately and rapid identification of some types of PVA in order to decrease the rate of mechanical ventilation-related adverse events.

General comments:

-The  studied  algorithm has a high specificity, PPV and NPV (better for the double trigger algorithm), but a lower sensitivity. Maybe the authors can be more specific about the "potential for large-scale clinical application".

-The authors mentioned that the accuracy of diagnosis is limited to specific types of PVA (double triggering and ineffective triggering). For the insufficient inspiratory flow rate (a type of PVA included in the overall analysis), with accuracy and area under the ROC 89% and respectively, 0.84 maybe a comment may be included in the discussion.

-About the text editing :  for the line 74 and the line 320 a reformulation  may be appropriate

Author Response

This article promotes a network system with an automatic recognition algorithm that improve the accurately and rapid identification of some types of PVA in order to decrease the rate of mechanical ventilation-related adverse events.

General comments:

-The  studied  algorithm has a high specificity, PPV and NPV (better for the double trigger algorithm), but a lower sensitivity. Maybe the authors can be more specific about the "potential for large-scale clinical application".

Response:

Thank you very much for your comments.

It will lead to some seriously mistakes or cause the alarm fatigue of the medical staff due to the sensitivity is too high. Therefore, we use more specific alarms to highlight the reliability of each alarm.

This article focuses on developing a platform, building a model, and then conducting small-scale verification. Although the number of clinical patients included is not large, the total time is long enough. Therefore, it is sufficient from the data used for model validation. Of course, in the follow-up research, we planned to continue to study the relationship between the safety and effectiveness of the single-center clinical verification system and the multi-center automatic human-machine asynchronous pattern recognition and patient prognosis, so as to lay a more solid foundation for clinical promotion.

-The authors mentioned that the accuracy of diagnosis is limited to specific types of PVA (double triggering and ineffective triggering). For the insufficient inspiratory flow rate (a type of PVA included in the overall analysis), with accuracy and area under the ROC 89% and respectively, 0.84 maybe a comment may be included in the discussion.

Response:

Thank you very much for your comments.

We have added the discussion as “Through this technology, a set of data is analyzed, and the characteristics of each breath are used to determine whether there are the following two types of PVA: double triggering, ineffective triggering. Then, we iteratively developed a machine-learning model to classify each breath as normal or asynchrony. The overall accuracy of the above two types of PVA identification was 91%, and the area under the ROC curve was 0.92. The accuracy and area under the ROC curve were 97% and 0.95, and 79% and 0.80 for the double triggering and ineffective triggering. In the verification experiment, the accuracy and ROC of double triggering and ineffective triggering have achieved the expected results.”

In this article, the experimental verification does not involve insufficient inspiratory flow rate and early cycling, so the relevant content involving iinsufficient inspiratory flow rate and early cycling is hidden. With the deepening of the research, the team will explore the problems of insufficient inspiratory flow rate and early cycling.

-About the text editing :  for the line 74 and the line 320 a reformulation  may be appropriate

73  The Remote-VentilateView platform can summarize, classify, organize

74  and store data for subsequent event alarms, data analysis reports, data visualization and

75  other data applications.

Response:

Thank you very much for your comments.

We have revised and updated as “The Remote-VentilateView platform can store data and perform data classification, data analysis and other processing, providing data support for event alarms, data analysis reports, data visualization and other data applications.”

319 However, these could not be clinical routine. Therefore, a new method to solve

320 this problem is urgently needed.

Response:

Thank you very much for your comments.

We have revised and updated as “This study proposes to build a Remote-VentilateView platform to solve various PVA problems. This platform can partly solve the needs of doctors in critical care medicine for the use of mechanical ventilation, and lay the foundation for large-scale clinical application in the later stage.”
